# Research on Image Processing Resource Reconstruction Based on Load Balancing Strategy

**Yuxiao Deng ¹, Jingyu Liu ¹,\* and Yang Zhou ²**

1. Beijing Institute of Control and Electronic Technology, Beijing 102300, China; deng_yuxiao@163.com
2. School of Mechanical Engineering and Automation, Beihang University, Beijing 100191, China; zhouyango@buaa.edu.cn
* Correspondence: liu_jingyu@163.com

**Abstract:** With the development of intelligent vehicles, the vehicle image processing system has put forward increasing demand for computing resource utilization efficiency and real-time processing. However, the traditional information processing method that binds software and hardware severely restricts the efficient use of image processing resources. In order to solve this problem, this paper proposes a resource reconstruction scheme based on a load balancing strategy, which can realize unified management and dynamic allocation of image processing resources by establishing a system resource view. Then, this paper builds a physical verification platform and constructs a comparative verification experiment to verify the effectiveness of the resource reconstruction scheme. Experimental results prove that this resource reconstruction scheme can effectively improve the resource utilization efficiency of multi-core digital signal processing (DSP), realize software-defined hardware functions, and optimize the real-time performance of parallel processing.

**Keywords:** resource reconstruction; multi-core digital signal processing; load balancing; task scheduling algorithm

## 1. Introduction

In recent years, with the development of the intelligence degree of intelligent vehicles, the on-board control system has required a higher environmental perception ability and a multi-source information processing ability, where the high-resolution image information processing ability is determined by the utilization efficiency of processing resources. To achieve unified system-distributed computing resource management, we optimize the architecture of the vehicle control system and propose an image processing resource reconstruction method.

The vehicle control system architecture has experienced long-term development. Cheng J. [1] proposed a truck control system based on a controller area network (CAN) bus in 2001, which successfully realized the data sharing of an ABS braking device and improved the performance of an ABS system. However, the bandwidth of the CAN bus is narrow, and the real-time data interaction cannot be guaranteed . In order to achieve real-time data interaction, Zhu S.Y. [2] designed a vehicle control system based on an Ethernet bus for the single-soldier unmanned tracked vehicle and successfully used Ethernet in series with Global Positioning System (GPS), ultrasonic sensors, and an internal control system to achieve intelligent obstacle avoidance for the unmanned tracked vehicle. However, since the data transmission bandwidth is in kilobytes (KB), if any image processing equipment is introduced, the system cannot meet the real-time transmission on the megabyte (MB) or gigabyte (GB) level . Huang Lei [3] designed an intelligent networked vehicle hybrid control system based on the time-sensitive networking (TSN) network, which realized the intelligent tracking of driverless vehicles, but the TSN network was only used for audio and video transmission in the system, without too much participation in data processing, and the application scenarios can be extended.

Early PC microcomputers did not have simulation systems, so the vehicle control system used two sets of PC microcomputer systems in order to constitute a closed loop of the system; however, this method is too complex, and the design cost is too high [4]. Chen C.D. [5] used DSP for intelligent image processing of unmanned vehicles and successfully transplanted the road environment detection algorithm to DSP but did not design for DSP's real-time data processing. In order to improve the system's flexibility, Bi J.Q. [6] designed a multi-level vehicle control system, splitting up the control system into four layers, including a sensor layer, a perception layer, a decision-making layer, and an executive layer, which realized the intelligent perceptual decision-making system, effectively improved system's flexibility, and was equipped with algorithm independence, providing ideas for the design of reconfigurable system software . In the existing research related to DSP reconstruction, some studies do not focus on DSP itself but combine a field-programmable gate array (FPGA) and DSP and use the programmable characteristics of FPGA to indirectly realize DSP reconstruction [7–10]. This method, however, cannot be applied to a control system that only uses DSP. In order to improve the resource utilization efficiency of multi-core DSP under the constraint of low cost, Zhao Q. [11] proposed an airborne reconfigurable control system based on the parallel connection of multiple DSPs. This had both reconstruction ability and strong scalability, but it needed FPGA as auxiliary, and the system's structure is relatively complex, and the real-time reconfiguration needed to be strengthened. Chen [12] proposed a spaceborne imaging algorithm based on multi-core DSP, which evenly distributes the algorithm to eight cores for parallel processing. However, this allocation method does not conduct online evaluation of core availability, which is blind. By building a dynamic library, Wang G.X. [13] proposed a reconstruction method using a DSP dynamic link library and realized dynamic loading of files, However, this could not change the system software structure, and the scheme was less flexible. Based on a dynamic software update, He S.Z. [14] proposed a DSP reconstruction method and realized the dynamic reconfiguration between DSP cores, having good real-time performance and flexibility; however, the scheme did not consider cross-device reconfiguration, and there is still room for expansion. Song Y. [15] built a parallel processing platform by implanting an operating system in each core to achieve real-time parallel processing of data. However, the construction process of this method is relatively complicated and conflicts with the strong constraints on the resource space of multi-core DSP. To solve the load balancing problem [16–20], Wang H.B. [21] proposed an improved weighted minimum connection load balancing scheduling algorithm for web server clusters, which separately scheduled tasks according to resource requirements and effectively improved the resource utilization efficiency; however, there was no dynamic classification of tasks, and the scheduling effect can be optimized. Xia J.J. [22] proposed a parallel processing solution based on the round robin [23] method for multi-core DSP. Although it can improve load balancing, it does not conduct real-time assessment of the resource distribution status, which is still blind. Zhao H.L. [24] proposed a load balancing prediction algorithm for multi-core DSP, but it cannot achieve real-time updates of resource status distribution. Based on the minimum migration degree and segmentation degree, Huang S.J. [25] proposed a load balancing task scheduling algorithm that effectively reduced the excessive overhead and context switching times caused by task migration and improved the load balancing degree of an embedded control system. Based on Modified Ant Colony Optimization (MACO), Dai F. [26] proposed a task scheduling algorithm and realized optimal load balancing scheduling, but this scheme has only been tested by simulation.

In summary, the above-mentioned studies have carried out extensive research of a vehicle control system's resource allocation, but there are still shortcomings in improving the dynamic utilization rate of system resources. In order to improve system resource utilization, this study tries to realize dynamic reconstruction of image processing resources by scheduling task files across devices.

The study is organized as follows. In the next section, we introduce the architecture of the reconstruction control system. In Section 3, an image processing resource reconstruction

scheme is proposed, and a task scheduling algorithm based on load balancing is constructed. The comparative verification experiment and the experimental results are presented in Section 4 to demonstrate the performance of the resource reconstruction scheme. Finally, the discussions and conclusion are made in Sections 5 and 6, respectively.

## 2. Architecture Design of the Reconstruction Control System

### 2.1. Vehicle Control System Architecture

The intelligent vehicle control system includes a perception system, a decision module, controllers, and a high-speed information transmission system. The perception system includes cameras, radars, GPS, and other information contacts and is responsible for collecting information about the surrounding environment of the vehicle. The decision module generates driving decisions, such as path planning and risk aversion, according to the input of environmental parameters. The controllers generate movements according to the driving strategy, such as steering, electronic throttle, speed change, and braking. The high-speed information transmission system is responsible for information interaction between nodes, sensor image transmission, command data transmission, and so on.

Taking an electric vehicle as an example, as shown in Figure 1, the six cameras, three ultrasonic sensors, and three millimeter-microwave radars together constitute a sensor array. All these sensors are deployed around the vehicle to collect the surrounding environmental parameters. The collected environmental parameters are sent to the integrated information processing module through the high-speed information transmission system. The integrated information processing module contains multiple boards for image processing, driving decision generation, control command transmission, and others.

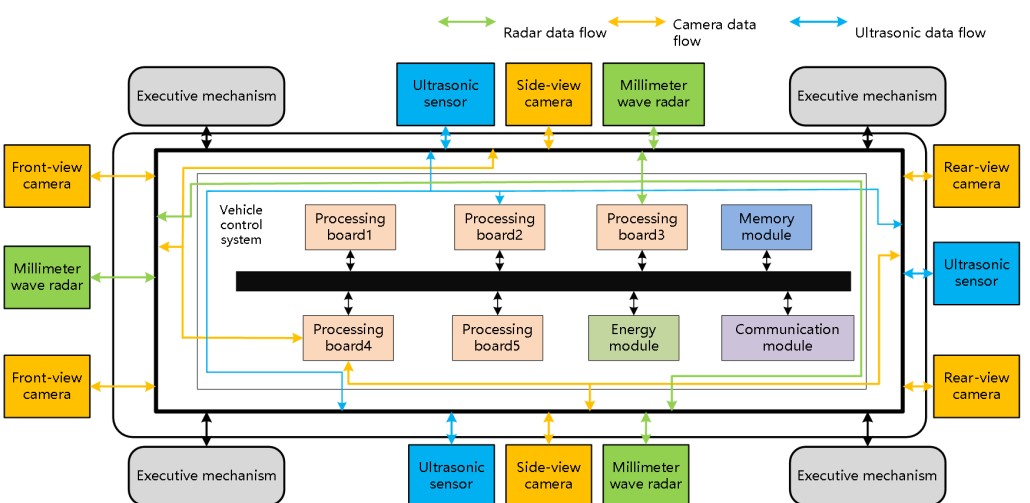

**Figure 1.** Schematic diagram of the vehicle control system architecture.

### 2.2. Hardware Architecture Design

To simplify the vehicle control system, the study focuses on the image processing resource reconstruction, and each processing unit is simplified to a TMS320C6678(C6678) DSP board, which has high performance, low energy consumption, and an operating frequency up to 10 GHz. It also integrates the high-speed TareNet bus with a communication rate of up to 2 TB/s. It supports a variety of serial ports and high-speed communication interfaces and has 20 GMAC/s or 160 GFLOP/s floating-point operation capability. In addition, each core of C6678 can run at a frequency of 1.25 GHz. All eight cores of C6678 can be reset independently, supporting independent access [27]. The internal architecture of C6678 is shown in Figure 2, and each DSP core has a two-layer memory architecture: Level 1 data memory (L1D), Level 1 program memory (L1P), and Level 2 memory (L2). All three can be configured as cache or independent memory (SRAM). In addition, this type of

DSP also has a 4 MB multicore shared memory controller (MSMC) and expandable DDR3 memory space.

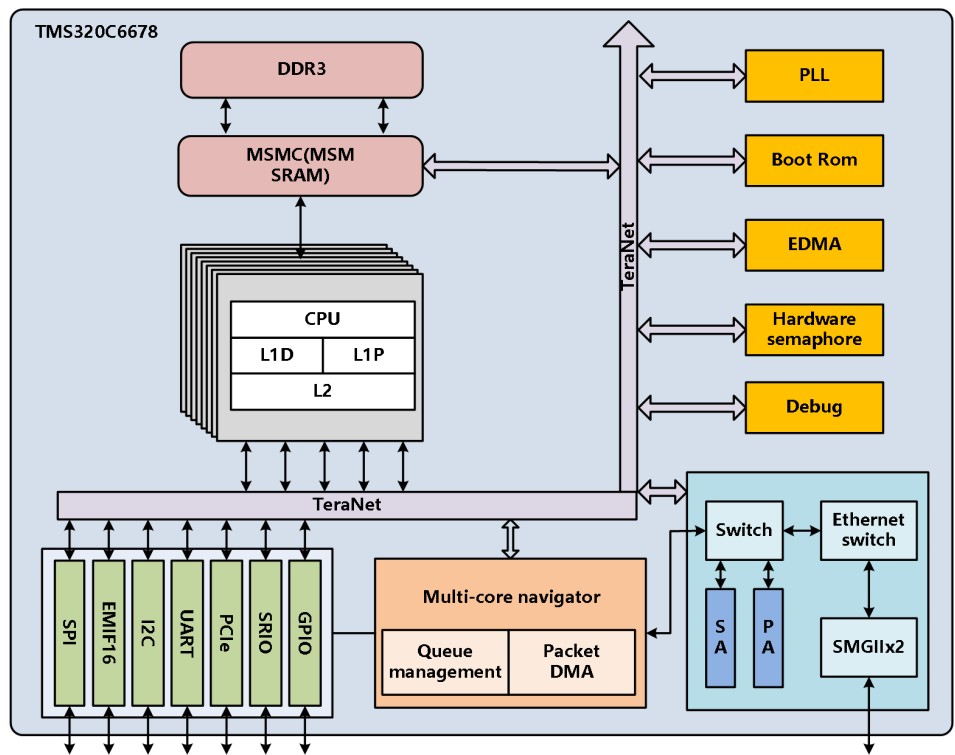

**Figure 2.** On-chip structure of TMS320C6678 DSP.

The hardware architecture of the control system is shown in Figure 3.

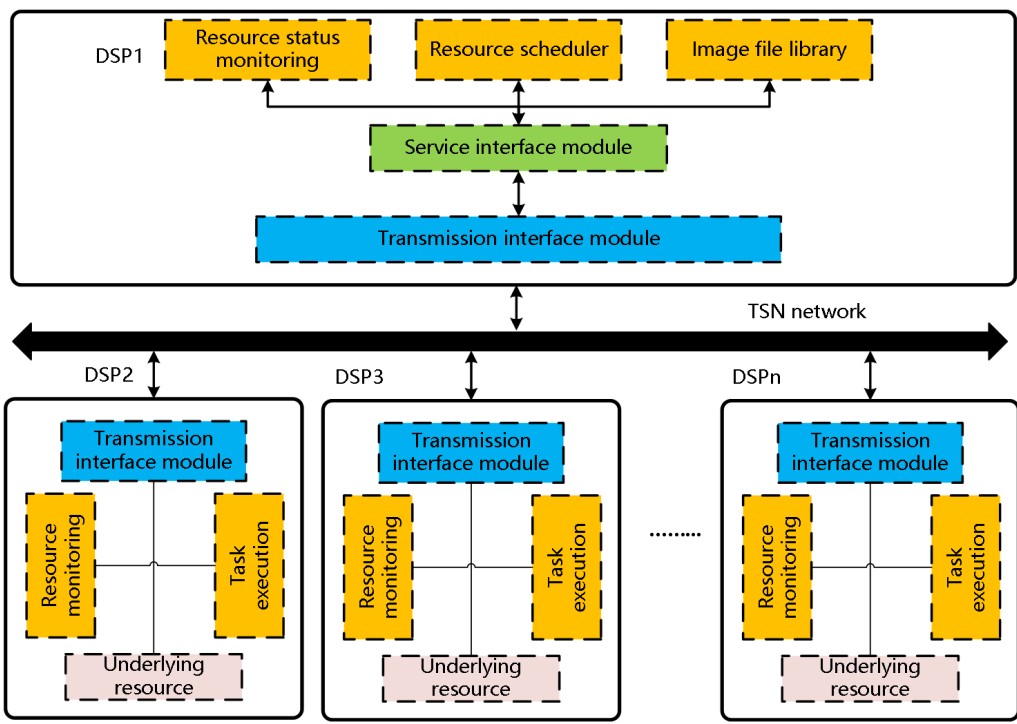

**Figure 3.** The hardware architecture of the control system.

In Figure 3, the hardware architecture of the control system adopts the master–slave mode, which is similar to the master–slave mode of C6678. All C6678 boards are mounted

on the TSN network bus. One C6678 is selected as the management node, which is responsible for the system commander, such as receiving data from sensors and software, generating the task deployment strategy and resource reconstruction strategy, scheduling the mirror files, and establishing the system state resource view. The rest of the C6678 boards act as work nodes, which are responsible for executing the instructions of the management node, processing images, and feeding back their own state resource parameters. The purpose of the transmission interface modules is to eliminate communication protocols and implement modular mounting. Each working node detects its own underlying resources, periodically packages the resource data, and transmits them to the system through the TSN network. The management node collects and generates real-time status resource views. In addition, when reconstruction is triggered, the management nodes generate reconstruction scheduling policies based on the resource views. The resource scheduler extracts files from the image file library and deploys them to working nodes through the TSN network. The working nodes execute reconstruction tasks and feed the results back to the management node. The basic process of reconstruction is shown in Figure 4.

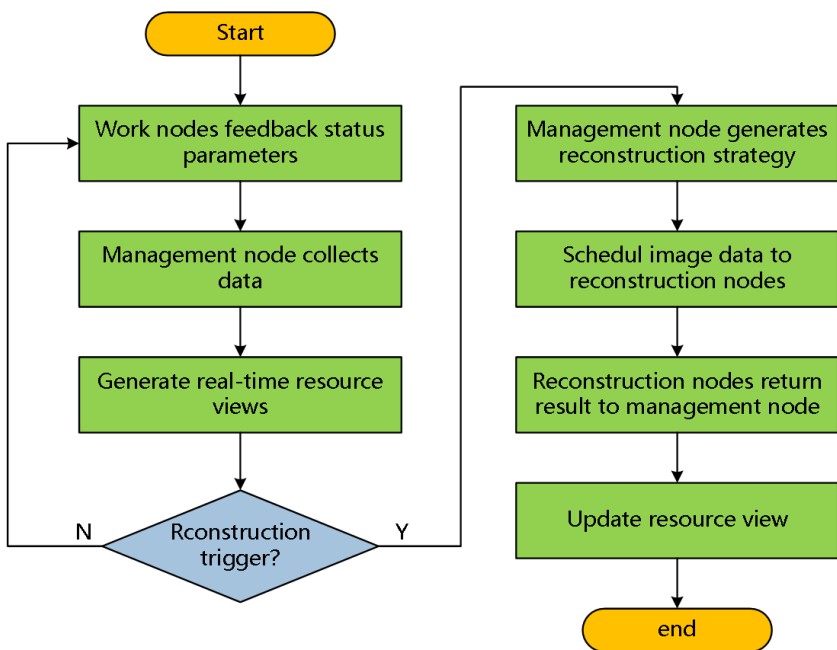

**Figure 4.** Basic process of reconstruction.

### 2.3. Software Architecture Design

To realize resource reconstruction, the study decouples tasks from the processing resources and breaks the binding between hardware devices and software functions. Additionally, the software architecture of the control system is designed hierarchically and divided into four layers, including a hardware resource layer, an operating system layer, a service interface layer, and a task requirement layer. The hardware resource layer is composed of multi-core DSP core processing resources such as NOR flash, static random-access memory (SRAM), Double Data Rate 3 Synchronous Dynamic RAM (DDR3), and TSN network communication resources. The operating system layer is the sys/bios embedded operating system, which provides a variety of threading modes for the use of programs. It can also freely configure the DSP memory addressing space through the express DSP Components-tools (XDC-tools) software driver, make schedulable program files, and configure API interfaces. Then, the service interface layer includes various APP programs that make up the task flow of the control system, such as algorithm programs, drivers, human–computer interaction architecture, databases, and resource status acquisition. The service interface layer also includes various protocol interfaces, such as communication protocols, interface files, a task scheduler, and a data messaging interface. Finally, the

perceptual information is the input data of the task requirement layer, which is collected by the sensor array, and the output of the layer is the task flow [28]. The software architecture design is shown in Figure 5.

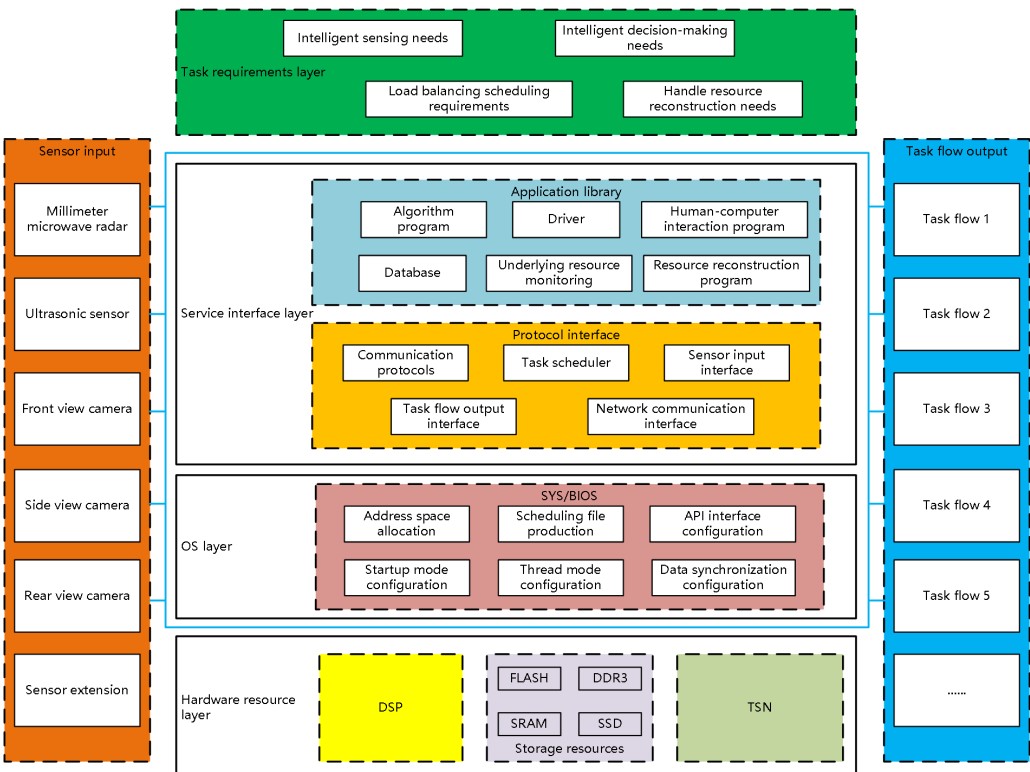

**Figure 5.** The software architecture of the control system.

In Figure 5, the software organization structure unbinds software programs and hardware resources, allowing users to perform customized operations according to task requirements. It can also coordinate and manage control system resources, laying a good foundation for the reconstruction of image processing resources.

## 3. Image Processing Resource Reconstruction Scheme Based on Load Balancing Strategy

When the DSP performs complex operations, the occupancy rate of the computing resources, the bus resources, and the storage resources increases sharply, causing DSP performance to decrease. In order to solve this problem, the study designs a load balancing reconstruction scheme, which contains two aspects: a scheduling algorithm based on the load balancing strategy and a reconfiguration scheme.

### 3.1. Scheduling Algorithm Based on Load Balancing Strategy

In order to improve reconstruction efficiency, the study proposes a reconstruction scheduling algorithm based on the load balancing strategy. The algorithm first builds a system resource view, which monitors the running status of each core, currently running task types and heartbeats in real time. When image data are received, the algorithm checks the real-time monitoring results of the resource view, quickly selects light-load nodes by comparing with the load balancing threshold and constructs a scheduling resource pool vector, and then dynamically allocates the image to the idle cores in the scheduling resource pool to deal with. This section will elaborate on the reconstruction scheduling algorithm based on the load balancing strategy from three aspects: the resource view establishing method, the load balancing evaluation method, and the scheduling algorithm scheme.

### 3.1.1. Resource View Establishing Method

The DSP adopts the master–slave working mode internally. Each core of the DSP runs the resource collection interface function UpdateStatusData (u32 taskNum). The input parameter of the function UpdateStatusData (u32 taskNum) is the core serial number. Then, each core builds a resource parameter array, including node number, running status, heartbeat, and task number. Each work core sends a resource parameter to core 0 during running time. Core 0 collects the resource parameters of the other cores and sends data packets to the management core. The management core stores the resource status parameters of the entire system in multiple structStatusFeedBack structures and then builds a system resource status parameter view.

### 3.1.2. Load Balancing Evaluation Method

Task scheduling must examine the load balance of all nodes in each period. If a core has too many unfinished tasks, then it is not suitable to be assigned new tasks because it requires a very long time. Therefore, when the system performs load balancing scheduling, it first needs to form a scheduling resource pool based on the resource view. We define the load parameter $L$, which describes the number of images scheduled for a single core within a certain period. Then, we define parameter $M$, which describes the number of existing image processing tasks on the core at the current moment. The weighted sum of these two parameters gives the core comprehensive performance index $Q$, as shown in Equation (1):

$$Q = \alpha_1 L + \alpha_2 M. \tag{1}$$

In Equation (1), $Q$ represents the comprehensive performance index of the core; $\alpha_1$ and $\alpha_2$ represent the weight parameters, respectively; and $Q_t$ is set as a threshold such that, when $Q < Q_t$, the core will be included in the scheduling resource pool. Then, we can schedule the unprocessed data for cores in the scheduling resource pool.

A scheduling resource pool array $P$ at the first scheduling time $t_0$ is shown in Equation (2):

$$P = [P_1, P_2, P_3, \ldots \ldots P_n]. \tag{2}$$

In Equation (2), the scheduling resource pool array $P$ includes all schedulable cores. If the number of existing image processing tasks on the core $i$ is zero at the current moment $t_0$, then $M_i = 0$, $P_i = 1$. On the other hand, if the number of existing image processing tasks on the core $i$ is not zero at the current moment $t_0$, then $P_i = 0$.

According to Equations (1) and (2), the core status information of the scheduling resource pool is updated in real time, and then the images will be allocated to idle cores for processing. In practical applications, the task scheduling scheme also needs to consider the relationship between the resource view establishment time $T_A$, the image deployment time $T_B$, and the image processing time $T_C$, as shown in Equation (3):

$$T_C \gg T_A + T_B. \tag{3}$$

In Equation (3), $T_C$ is much longer than $T_A + T_B$. Therefore, we can ignore the impact of $T_A$ and $T_B$.

### 3.1.3. Scheduling Algorithm Scheme

This paper proposes an image processing resource reconstruction and scheduling algorithm based on the load balancing evaluation method proposed above. At $t_0$, there are $N$ images that need to be processed in parallel. Then, the scheduling algorithm is triggered, and the system allocates images to the core in the scheduling resource pool with $P_i = 1$. After $T_0$, these allocated images are processed, and then the scheduling resource pool array $P$ is updated at the same time. Repeating the above operation $K$ times, all $N$ images are processed, the scheduling algorithm is ended, and then the system gets the time $T$ for parallel processing, as shown in Equation (4):

$$T = K \times T_0, \tag{4}$$

where the steps of the scheduling algorithm are as follows:

1.  There are $N$ images that need to be processed in parallel, and the system calculates the comprehensive performance index $Q$ of each core. If $Q < Q_t$, the core will be included in the scheduling resource pool, and then the scheduling resource pool is built;

2.  At $t_0$, the control system checks the scheduling resource pool array $P$. If the number of elements is $S$ in $P$, with the values equal to $P_i = 1$, then the number of cores that can participate in parallel computing is $S$;

3.  The $S$ images will be sent to these cores that can participate in parallel computing. Then, the number of images is updated to $N' = N - S$, which need to be processed in parallel. If $N' \leq 0$, the algorithm goes to step 5; otherwise, it goes to step 4;

4.  Then, update the scheduling resource pool array $P$ in real time, and we can get a new value of $S$ and repeat step 3;

5.  According to the number of times $K$ needed to repeat step 3, we can get the total time $T = K \times T_0$ of the parallel computing, and the scheduling algorithm is over.

The scheduling algorithm flow chart is shown in Figure 6, and the timeline chart of the scheduling algorithm is shown in Figure 7.

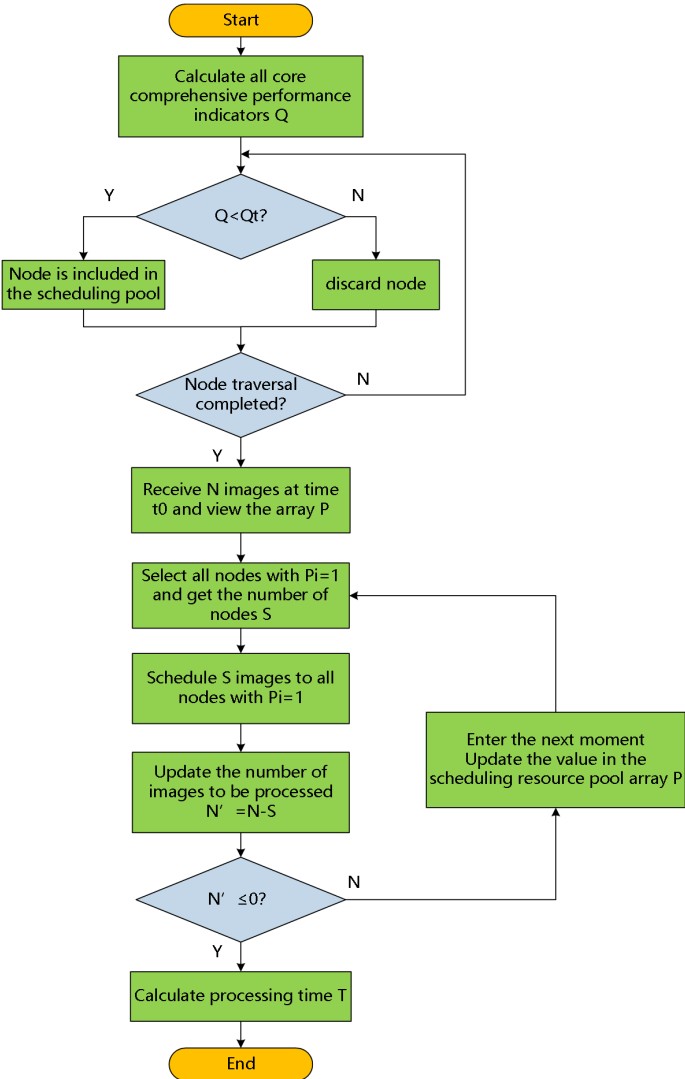

**Figure 6.** Scheduling algorithm process.

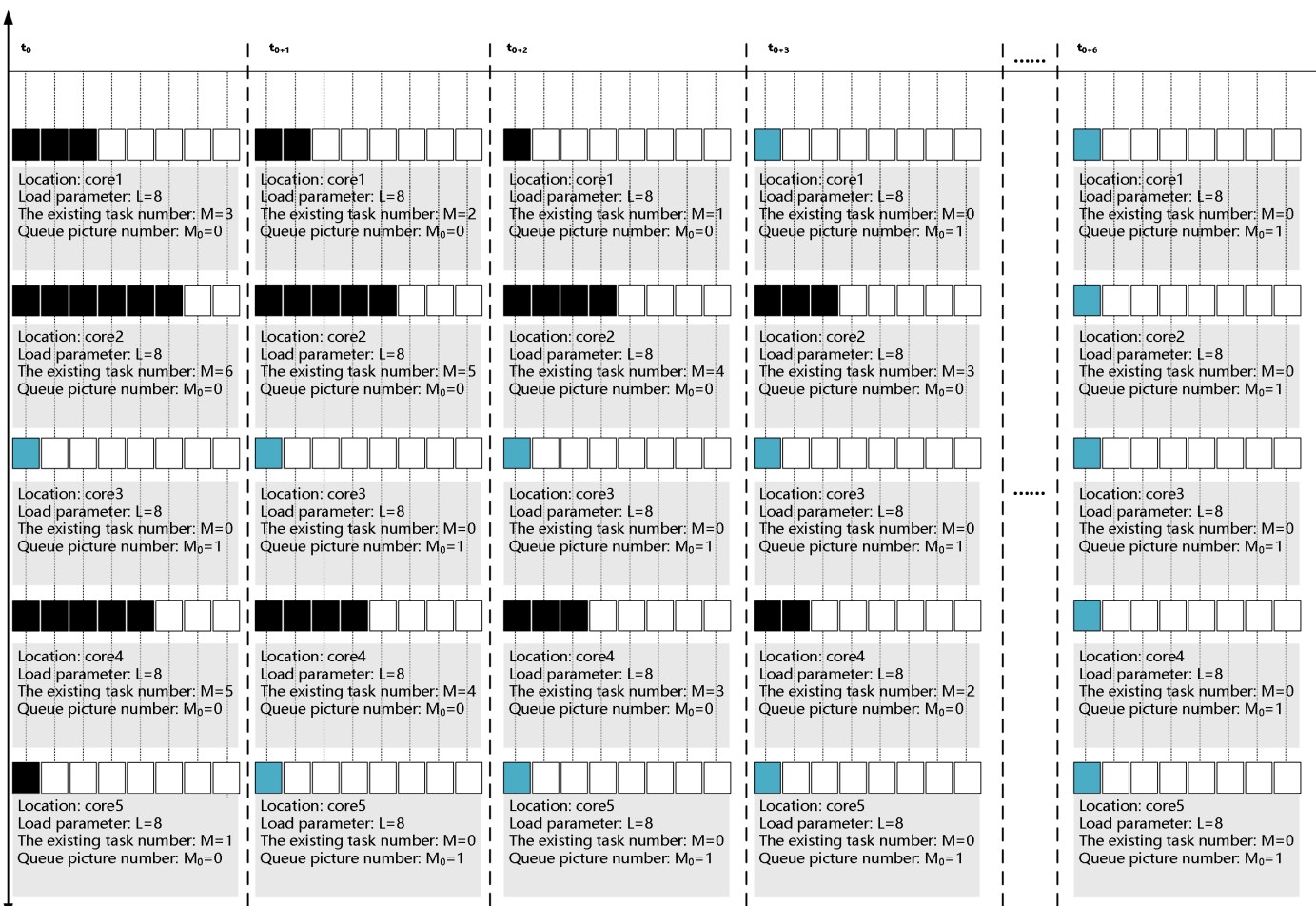

**Figure 7.** Scheduling algorithm process.

### 3.2. Reconstruction Scheme

The image processing task is an event-triggered task and does not generate status data. Therefore, the code of the task can be made into a mirror file and independently stored and scheduled under the C6678 programming system. The mirror files can be scheduled at high speed between different DSP nodes. The following reconstruction scheme will be described from three aspects: mirror file production, data package format provisions, and image processing resource reconstruction process.

#### 3.2.1. Mirror File Production

The control system makes mirror files based on master–slave DSP architecture for all cores respectively, as shown in Figure 8, where core x represents the DSP cores, such as core 0, core 1, core 2, core 3, core 4, core 5, core 6, and core 7. The function of core 0 is startup, resetting, and loading image processing files for other cores. After compiling, the core x md-parameter file and the core x out-format file will be generated. The code segment and data segment are added to the out-format files, where the parameters include startup mode, large or small end mode, and phase-locked loop (PLL) frequency. Then, the core x md-parameter file and the core x out-format file will be transmitted into a btbl-format file by Hex6x.exe. Finally, the btbl-format file will be transmitted into a bin-format file by Txt2bin-tool, where the bin-format file is the mirror file and is stored in a physical storage unit, such as NOR flash or DDR3.

The structure of the mirror file stored in the slave core contains just the program entry address, code segment, and data segment, as shown in Figure 9.

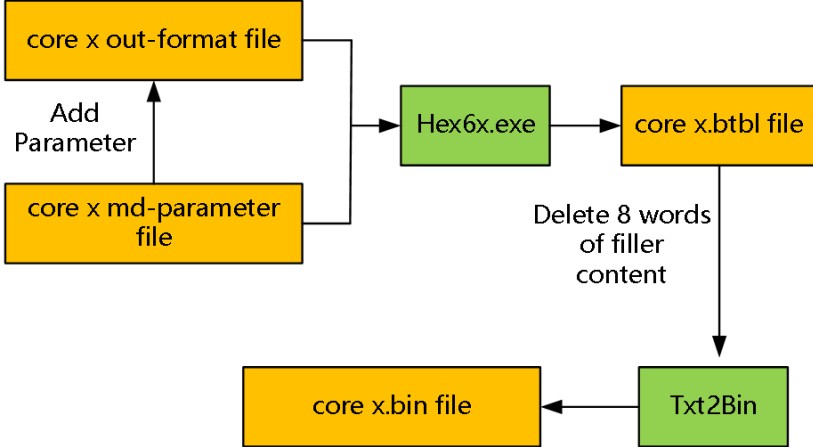

**Figure 8.** Production process of mirror files.

| _c_int00 address | one word |
| --- | --- |
| .text length | one word |
| .text origin address | one word |
| .text content | It depends |
| .cinit length | eight words |
| .cinit origin address | eight words |
| .cinit content | It depends |
| ...... | ...... |
| 0 pending | |

**Figure 9.** Mirror file structure.

### 3.2.2. Data Package Format Provisions

The data packet format is designed based on TCP/IP protocol. The packet is divided into five parts, including the Synchronization Header, Device Word, Data Length, Effective Data, and Parity Bit. The length of the Synchronization Header (0xEB) is 1 byte, the length of Device Word is 1 byte, the length of Data Length is 2 bytes, but the Effective Data length depends on the actual situation. Finally, the length of Parity Bit is 4 bytes. The data package format is shown in Figure 10.

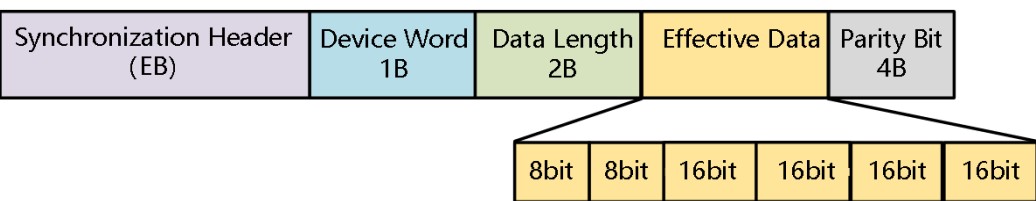

**Figure 10.** The data package format.

### 3.2.3. Image Processing Resource Reconstruction Process

The hardware architecture of the control system shown in Figure 1 can be simplified into a distributed computing architecture that is composed of five DSP nodes (DSP1, DSP2, DSP3, DSP4, and DSP5), in which core 0 of DSP1 is designated as the management core, and core 1 to core 7 of each DSP node are used as the working cores. The management core monitors the resource status of the distributed computing architecture in real time. When the distributed computing architecture receives the parallel processing task requirements, the management core formulates a reconstruction strategy based on the monitoring results

of the working cores, sends the reconstruction command to the reconstruction node, and then the management core schedules the mirror file from the physical storage unit to the reconstruction cores. The reconstruction process is shown in Figure 11.

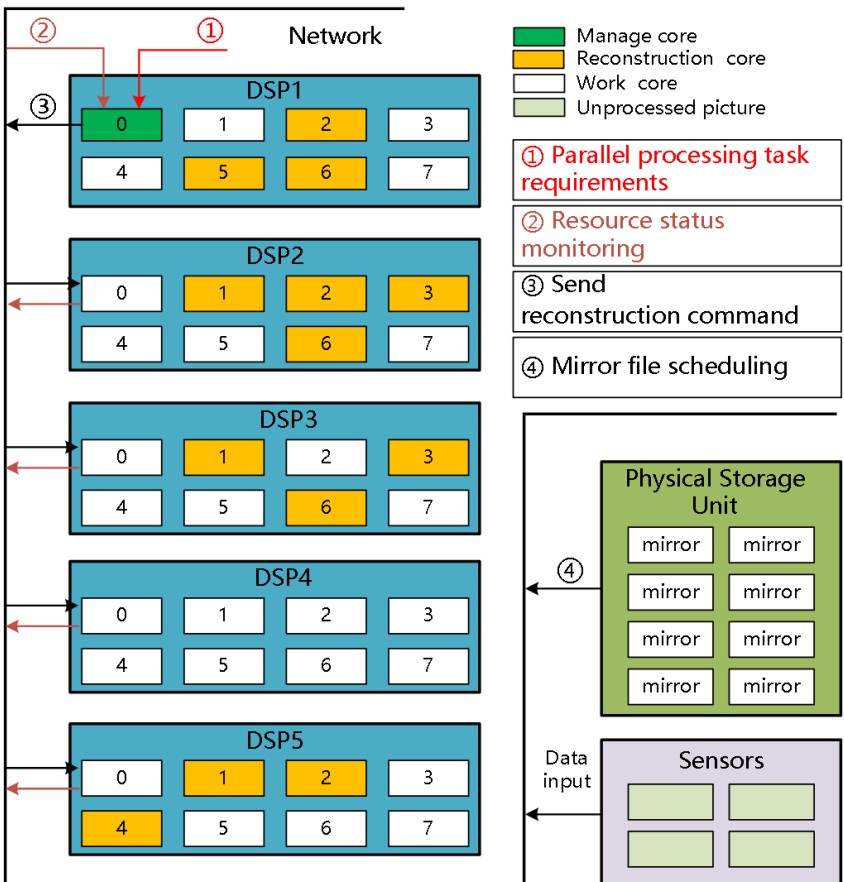

**Figure 11.** Image processing reconstruction process.

## 4. Experiment

### 4.1. Experiment for System Environment

The focus of this experiment is to verify the image processing ability of the system environment shown in Figure 1. The high-speed camera records five groups of pictures in different sizes continuously, where each group contains 10 frames, and the record period of each frame is 100 μs. Then, these five groups of images will be processed by the edge detection algorithm in one working core respectively, where the five groups of images are shown in Figure 12, and the processing time of each group is shown in Table 1.

According to Table 1, the average processing time $T_a$ grows with the increase in the image size.

**Table 1.** The processing time of each group.

| Core Number | $t_0$ | $t_1$ | $t_2$ | $t_3$ | $t_4$ | $t_5$ | $t_6$ | $t_7$ | $t_8$ | $t_9$ | $T_a$ |
|---|---|---|---|---|---|---|---|---|---|---|---|
| Core 1 | 9 ms | 9 ms | 9 ms | 9 ms | 9 ms | 9 ms | 9 ms | 9 ms | 9 ms | 9 ms | 9 ms |
| Core 2 | 19 ms | 19 ms | 18 ms | 19 ms | 19 ms | 18 ms | 19 ms | 19 ms | 18ms | 18 ms | 18.6 ms |
| Core 3 | 52 ms | 52 ms | 52 ms | 51 ms | 53 ms | 52 ms | 52 ms | 53 ms | 53 ms | 53 ms | 52.3 ms |
| Core 4 | 74 ms | 74 ms | 75 ms | 75 ms | 74 ms | 75 ms | 75 ms | 74 ms | 74 ms | 74 ms | 74.4 ms |
| Core 5 | 96 ms | 95 ms | 94 ms | 94 ms | 94 ms | 96 ms | 95 ms | 96 ms | 95 ms | 95 ms | 95 ms |

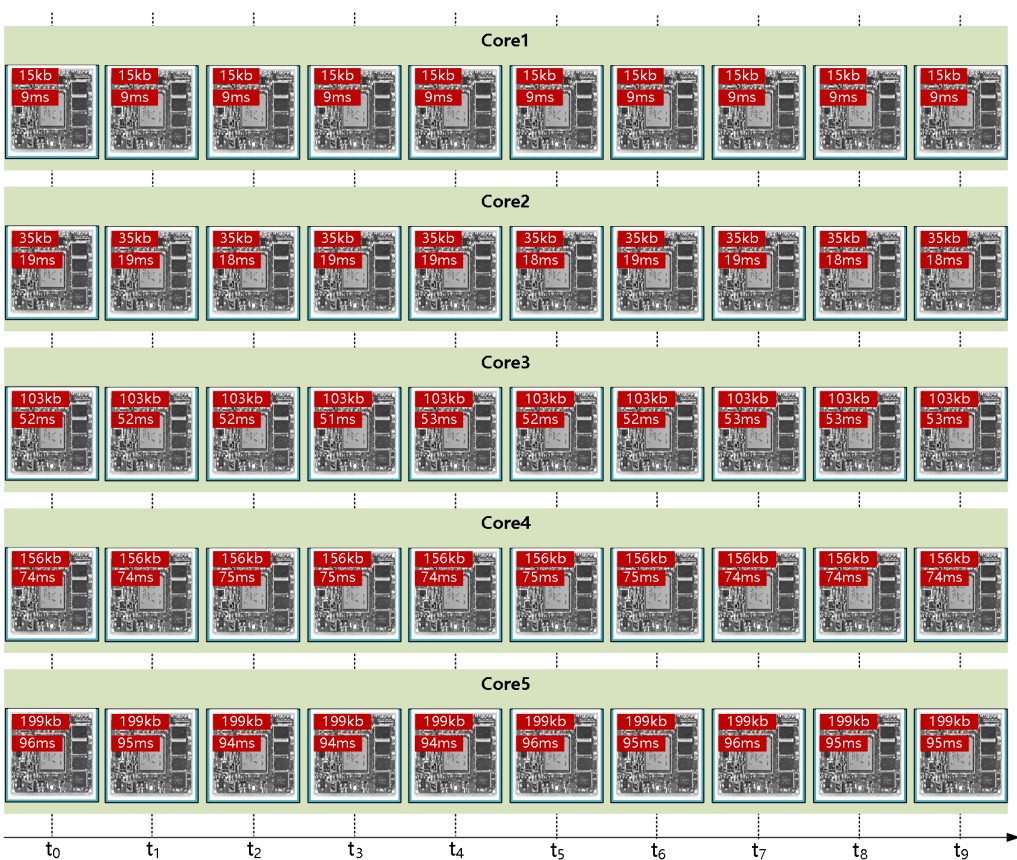

**Figure 12.** Five groups of images.

## 4.2. Resource Reconstruction Experiment Based on the Traditional Method

The focus of this experiment is to verify the efficiency of the traditional resource reconstruction method for the parallel processing task. The high-speed camera records 50 frames of pictures continuously, where the size of each frame is 103 KB, and the record period of each frame is 100 μs. These 50 images will be equally divided into five groups, and then these five groups of images will be processed by the edge detection algorithm. Resource reconstruction based on the traditional method is shown in Figure 13.

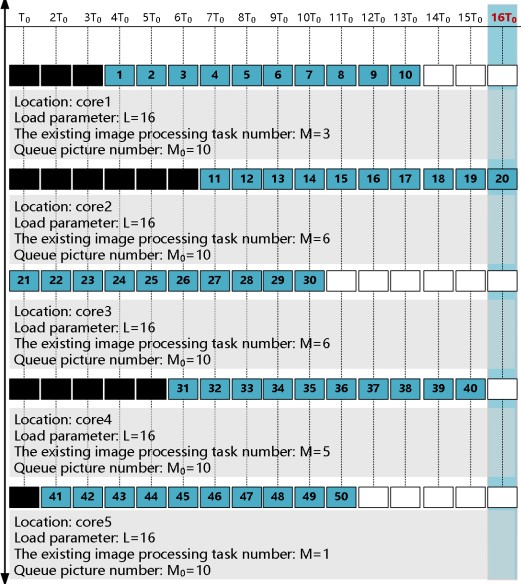

**Figure 13.** Resource reconstruction based on the traditional method.

The processing time of each core for the parallel processing task in the traditional method is shown in Table 2.

**Table 2.** The processing time of each core in the traditional method.

| Core Number | Total Number of $T_0$ | Time |
|:---:|:---:|:---:|
| Core 1 | 13 | 676 ms |
| Core 2 | 16 | 832 ms |
| Core 3 | 10 | 520 ms |
| Core 4 | 15 | 780 ms |
| Core 5 | 11 | 572 ms |

According to Figure 13 and Table 2, the usage efficiency of the working cores is relatively low because some cores need to wait for the existing image processing task to end before they can start processing the current task. Therefore, the traditional method seriously limits the real-time performance of tasks and resource utilization efficiency.

### 4.3. Resource Reconstruction Experiment Based on Load Balancing Strategy

The focus of this experiment is to verify the efficiency of the resource reconstruction scheme based on the load balancing strategy for the parallel processing task. The high-speed camera records 50 frames of pictures continuously, where the size of each frame is 103 KB, and the record period of each frame is 100 μs. Then, these 50 images will be processed by the edge detection algorithm. The resource reconstruction based on the load balancing strategy is shown in Figure 14.

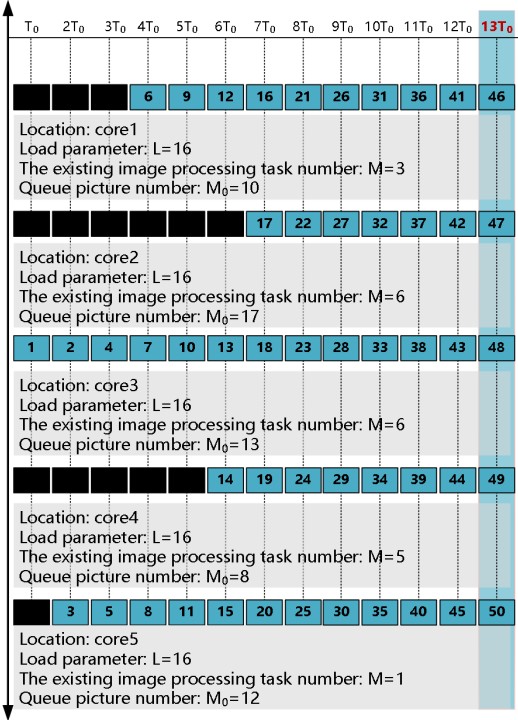

**Figure 14.** Resource reconstruction based on the load balancing strategy.

The processing time of each core for the parallel processing task based on the load balancing strategy is shown in Table 3.

According to Tables 2 and 3, compared with traditional methods, the processing time for the parallel processing task based on the load balancing strategy is shorter and is optimized by 18.75%. Moreover, the computing power of each core is fully utilized.

**Table 3.** The processing time of each core in the resource reconstruction scheme.

| Core Number | Total Number of $T_0$ | Time |
|:---:|:---:|:---:|
| Core 1 | 13 | 676 ms |
| Core 2 | 13 | 676 ms |
| Core 3 | 13 | 676 ms |
| Core 4 | 13 | 676 ms |
| Core 5 | 13 | 676 ms |

## 5. Discussions

According to the experiment results in Section 4, the resource reconstruction scheme based on the load balancing strategy reduces the image processing time and improves resource utilization efficiency. The advantages of the reconstruction scheme can be summarized as follows:

1.  The reconstruction scheme lift the binding of software tasks and hardware devices;
2.  The reconstruction scheme achieved unified visual management of system processing resources by inserting interface functions into tasks;
3.  By scheduling mirror files across nodes and using the reconstruction scheduling algorithm based on the load balancing strategy, the reconstruction scheme reduced the image processing time and improved resource utilization efficiency.

Compared with the method in Ref. [15], the method proposed in this article can realize resource reconstruction based on load balancing without deploying an operating system on each core and has advantages in development difficulty and code writing complexity. Compared with the methods in Refs. [12,22,23], the method proposed in this article can conduct real-time assessment of resource distribution and avoid blindness in resource allocation. However, this paper does not consider the reconstruction of heterogeneous resources, nor does it verify the reconstruction effect of GB-level or larger-capacity images. Therefore, the next stage of our research is to verify the feasibility of this reconstruction scheme in heterogeneous resources, and then realize the image processing resource reconstruction for large-scale images.

## 6. Conclusions

In this paper, we propose an image processing resource reconstruction scheme based on the load balancing strategy, which can reduce the image processing time and improve resource utilization efficiency for parallel processing tasks. The comparative experiments based on the physical platform verified the effectiveness of the reconstruction scheme based on the load balancing strategy. The next step of this research will focus on how to dynamically allocate resources to tasks based on the load balancing strategy when there are multi-dimensional coupling constraints between parallel tasks.

**Author Contributions:** Y.D. implemented the experiment and wrote the first draft of the paper, J.L. and Y.Z. provided funding for the paper and revised it. All authors have read and agreed to the published version of the manuscript.

**Funding:** This research received no external funding.

**Data Availability Statement:** The data presented in this study are available in this article.

**Conflicts of Interest:** The authors declare no conflicts of interest.

## Abbreviations

The following abbreviations are used in this manuscript:

| | |
|---|---|
| DSP | Digital Signal Processing |
| CAN | Controller Area Network |
| GPS | Global Positioning System |
| KB | Kilobyte |

| MB | Megabyte |
|---|---|
| GB | Gigabyte |
| TSN | Time-Sensitive Networking |
| FPGA | Field-Programmable Gate Array |
| MACO | Modified Ant Colony Optimization |
| PLL | Phase-Locked Loop |
| MSMC | Multicore Shared Memory Controller |
| SRAM | Static Random-Access Memory |
| DDR3 | Double Data Rate 3 |
| ASCII | American Standard Code for Information Interchange |
| XDC-tools | eXpress DSP Components-tools |

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
