# Peer review of "Research on Image Processing Resource Reconstruction Based on Load Balancing Strategy"

_electronics, doi:10.3390/electronics13061027_

Round 1
Reviewer 1 Report
Comments and Suggestions for Authors
The authors raise an important issue and precisely explained its purpose and assumptions. However, supplementing the article with several elements will increase its value and allow for better understanding by the reader.
1. It is worth adding a few sentences of description for figure 2.
2. In Section 3.1, it is worth describing in more detail scheduling algorithm based on load balancing strategy that was used.
3. In the discussion section, there is no comparison of the authors' results to other results published in the literature. The obtained results should be compared to the solutions of other authors.
4. The conclusions section lacks a plan for further work. Future direction can be added in a new paragraph of the conclusion part.
5. The scope of the literature is limited. It is worth adding a few recent items on the subject matter presented
Comments on the Quality of English LanguageThe language page is correct, minor corrections required.
Reviewer 2 Report
Comments and Suggestions for Authors
The authors provided a way of processing images to utilize the resources efficiently and get the outcome in real-time, which is crucial for vehicle use cases. The research work is timely and of prime importance. The following are comments that the authors should consider addressing.
1. The authors are advised to provide an in-depth comparison of their work with existing literature to establish its novelty.
2. Please correct Figure 1. Some boxes have a single letter on the second line.
3. kindly consider rewriting sentences that are not clear. Please see the comment on the quality of English.
Comments on the Quality of English LanguageThere are lines throughout the paper that need to be clarified. I request that those sections be rechecked and rewritten for clarification. For example, the following line in the abstract is hard to interpret: " In order to solve the problem of low efficiency of image processing system resources and the application software binding with hardware device."
Reviewer 3 Report
Comments and Suggestions for Authors§ What specific metrics or benchmarks were used to evaluate the effectiveness of the resource reconstruction scheme, and how do they correlate with real-world performance indicators?
§ Can the proposed scheme dynamically adapt to changes in workload characteristics and system requirements, and if so, how is this adaptability achieved?
§ How does the scheme handle potential bottlenecks or uneven distribution of processing tasks across different resources or cores?
§ What mechanisms are in place to ensure fairness and equitable resource allocation among competing processing tasks?
§ Are there any potential security implications or vulnerabilities introduced by the resource reconstruction scheme, particularly in terms of data privacy or system integrity?
§ How does the scheme address potential failure scenarios or unexpected disruptions in processing resources, such as hardware failures or network issues?
§ What considerations were given to the cost-effectiveness of implementing the resource reconstruction scheme, particularly in terms of hardware upgrades or additional software development efforts?
§ How does the proposed scheme compare to existing approaches or alternative strategies for improving resource utilization efficiency and real-time processing performance in image processing systems?
§ What level of transparency and configurability does the scheme offer to system administrators or developers in terms of fine-tuning parameters or adapting to specific use cases?
§ How do the findings from the comparative experiments and the demonstrated effectiveness of the resource reconstruction scheme contribute to the broader understanding of optimizing computing resource utilization and real-time processing performance in image processing systems, particularly in the context of intelligent vehicle applications?
§ Please, Adding the new contribution to the introduction section
Round 2
Reviewer 3 Report
Comments and Suggestions for Authors
No comments